# Spatial Threshold Effect of Industrial Land Use Efficiency on Industrial Carbon Emissions: A Case Study in China

**DOI:** 10.3390/ijerph18179368

**Published:** 2021-09-05

**Authors:** Xiao Zhang, Meng Li, Qiao Li, Yanan Wang, Wei Chen

**Affiliations:** College of Economics and Management, Northwest A&F University, Yangling 712100, China; 2157242347@nwafu.edu.cn (X.Z.); 15847108868@nwafu.edu.cn (M.L.); liqiao@nwafu.edu.cn (Q.L.); chen_wei@nwafu.edu.cn (W.C.)

**Keywords:** industrial carbon emissions, industrial land efficiency, panel threshold-model, spatial Durbin model

## Abstract

China’s industry is still in the middle of industrialization. Land use activities are crucial to the growth of carbon emissions. However, few scholars focus on the influence mechanism between industrial land use efficiency (ILUE) and industrial carbon emissions. In this paper, the threshold model and the spatial Durbin model are used to investigate the spatial threshold effect of industrial land use efficiency on industrial carbon emission from 2003 to 2018. The results show that ILUE of China’s provinces basically shows an improvement trend, with little difference in spatial distribution, showing a pattern of high in the eastern region and low in the western region. When economic development level (A) and technical level (T) are taken as the threshold variable, ILUE has a single threshold effect on industrial carbon emissions in the eastern region. In the central region, with a as the threshold variable, ILUE shows a double threshold effect on industrial carbon emission. Under the 0–1 geographical proximity weight matrix, the indirect spillover effect of ILUE on reducing regional carbon emissions is significant, and the indirect effect is even greater than that on regional carbon emissions. The spatial spillover effect is not significant in the eastern region. These findings have important practical significance for promoting regional industrial transformation and upgrading, optimizing land space and realizing high-quality economic development.

## 1. Introduction

The rapid industrialization and urbanization occurring in China has caused significant environmental pollution problems, particularly the emission of greenhouse gases, which have drawn the attention of scholars’ worldwide [1]. In 2006, China surpassed the United States to become the world’s largest carbon emitter and faces increasing pressure to reduce its carbon emissions. In order to do so, China promised to achieve a carbon peak by 2030 and carbon neutrality by 2060 as part of the Paris Agreement [2,3]. China’s economy is currently characterized by high growth, high investment, high energy consumption, and high carbon emissions. The total industrial output only accounts for approximately 35% of the GDP, while energy consumption accounts for nearly 70%, and carbon emissions exceeds 80% [4]. Land use activities are a major source of carbon emissions [5]. Intensive industrial land use helped to facilitate reductions in CO_2_ emissions. Improving the quality of land use is an important method for reducing environmental pollution [6,7]. Therefore, improving industrial land use efficiency (ILUE) is important for reducing industrial carbon emissions.

ILUE refers to the ability of regions or industries that allocate and use industrial land to obtain economic outputs [8]. The impact of ILUE on industrial carbon emissions varies depending upon the restraint mechanisms in place. The government’s strategies for improving ILUE also vary depending on the area’s state of economic and technological development. As shown in Figure 1, during the initial stage of economic development, the primary goal of the government is to pursue further economic growth. At the same time, due to factors such as a lack of capital, the government can only attract investment by selling large amounts of land at low prices. During this stage, the industrial structure is based primarily on heavy industry. Although industrial land use efficiency has been somewhat improved, the overall efficiency remains low. These high-energy consumption and high-carbon emitting industrial clusters further increase industrial carbon emissions. In the middle and late stages of economic development, the government’s pursuit of economic development slows down. At the same time, increases in industrial land use efficiency do not depend on the scale of the industry, but instead rely upon technological progress and upgrading the existing industrial structures. Industrial land use efficiency is continuously improved by the adoption of cleaner, more efficient production techniques as well as improvements to the industrial structures, all of which contributes to effectively curbing industrial carbon emissions.

The impact of industrial land use efficiency on industrial carbon emissions also varies depending on the area’s level of technological development. When the level of technological development is low, the main purpose of introducing further technology is to increase profits. This leads to a lack of environmental protection technology which causes serious problems with carbon emissions. Although the industrial land use efficiency is increased, industrial carbon emissions continue to rise. When the level of technological development improves further, the government pays more attention to investing in green and clean technologies, and promotes industrial upgrading through technological innovations which improve energy conservation [9]. Thus, improving ILUE can inhibit industrial carbon emissions.

Generally speaking, previous studies have focused on the calculation methods, influence factors and spatial pattern evolution of industrial land use efficiency. Few scholars focus on the influence mechanism between industrial land use efficiency and industrial carbon emissions. In fact, it is a complex that the relation between the two and there may have be nonlinear effects. When different regions are at different economic and technological levels, the impact of ILUE on industrial carbon emissions is vary. Therefore, the threshold model is implied to study the non-linear relationship between ILUE and industrial carbon emissions. Furthermore, the spatial spillover effect of ILUE on industrial carbon emission is deeply analyzed by using spatial Durbin model, and the regional differences of three regions in China are further discussed. This study aims to assess China’s industrial land use efficiency, investigate spatiotemporal distribution characteristics of industrial land use efficiency, and analyze the spatial threshold effect of industrial land use efficiency on industrial carbon emission. The conclusions have important reference value for optimizing industrial layout and realizing carbon emission reduction.

The remainder of this paper is organized as follows. Section 2 presents the literature review. Section 3 introduces the methods. Section 4 shows the data source. Section 5 reports the empirical results and discussion. Conclusions and policy suggestions are presented in Section 6.

## 2. Literature Review

With the growing urbanization and industrialization of China, land use plays an increasingly important role in carbon emissions [10]. Researcher have considered how to reduce carbon emissions by controlling the factors that influence land use. Land use primarily affects carbon emissions through changes in what the land is being used for and in what kinds of human activities occur on the land [11]. Extensive land use activities increase carbon emissions. Efforts to increase the economic output of land are the primary factor driving the growth of carbon emissions [12,13]. Industrial land use is the most important type of land use when considering carbon emissions and has attracted considerable research attention. The existing research mainly focuses on the efficiency of industrial land use, including measurement, influencing factors, spatial differences, and effects. However, there is no unified method for the calculation of ILUE. Some scholars choose economic indicators such as the total industrial output value and the industrial added value as output indicators to measure ILUE [14]. Other scholars believe that using economic benefits to express output is too one-sided. They take a more comprehensive approach, and consider environmental benefits and social benefits, among other aspects [15,16]. The primary factors that influence ILUE include the region’s level of economic development, industrial development, and the level of scientific and technological development [17,18,19,20]. In addition, land use efficiency can have spatial spillover effects; the improvement of land use efficiency in one region can have an impact on other regions through technological and industrial transfer [21,22].

With the continuous deterioration of the global climate, intensive land use has become a major research focus. Land use efficiency is widely considered a representative index of intensive land use. Li and Chen [23] studied Nanjing as an example case, and concluded that it was necessary to improve land use efficiency in order to reduce carbon emissions. Wang et al. [24] concluded that land use intensity had a positive impact on carbon emissions. Dong et al. [25] used the generalized method of moments (GMM) to show that there was a significant, dynamic correlation between urban land use efficiency and the intensity of carbon emissions. In the short term, urban land use efficiency had a driving negative effect on the intensity of carbon emissions. Wang et al. [26] analyzed the dynamic changes in the total factor carbon emission performances of industrial land in China, and found that the eastern region’s industrial land had better carbon emission performance than that of the central and western regions. Xie et al. [27] studied the relationship between carbon emissions and intensive industrial land use based on the STIRPAT (Stochastic Impacts by Regression on Population, Affluence, and Technology) model, and found that intensive industrial land use had a significant impact on carbon emission.

The research on industrial carbon emissions primarily focuses on the measurement of industrial carbon emissions, the factors influencing industrial carbon emissions, and the spatial nature of industrial carbon emissions. The existing literature on China’s industrial carbon emissions can be traced back to an early study by Shrestha and Timilsina [28], who analyzed the CO_2_ emission intensity of the power industry in 12 Asian countries, including China. The industrial output value was found to be the main factor driving the increase of industrial carbon emissions [29,30]. The region’s level of economic development and population factors also led to increases in carbon emissions. Reductions in energy intensity and the optimization of industrial structures were found to be key factors in reducing carbon emissions [31,32]. Industrial carbon emissions have both spatial correlations and spatial heterogeneity, and the spatial spillover effect should be considered [33,34].

The panel threshold approach has been widely used in many areas of applied economics and econometrics. Its basic principle is to select an economic parameter as the threshold variable, divide the regression sample into multiple intervals according to the selected threshold value, and then compare the changes of the influence coefficients of core economic parameters in different intervals [35,36]. Compared with traditional methods, the threshold model can test and estimate the existence of threshold effects according to the randomness of the exogenous variable grouping. At the same time, the samples are grouped endogenously according to the specific threshold. The threshold model can also repeatedly extract samples through the bootstrap method in order to improve the efficiency of significance testing [37]. At present, the threshold model is mainly used to test the nonlinear influencing factors of carbon emissions. Li et al. [38] used the panel model to study the impact of technological progress on carbon emissions. The study holds that when economic development exceeds a certain threshold, the impact of technological progress on carbon emissions changes from positive to negative. Similarly, Song [39] used the threshold model to analyze the impact of economic growth on carbon emissions. Wu et al. [40] discussed the impact of energy consumption on carbon emissions under different environmental regulations by using the dynamic threshold panel model. Zhang and Ma [41] used the extended dynamic threshold model to systematically analyze the relationship between industrial structures and the carbon emission intensity of 21 industrial sectors with different levels of economic development in eight developed countries from 1971 to 2014. Sun and Wang [42] used a threshold analysis to assess the impact of urbanization on carbon emissions.

## 3. Methods

### 3.1. Calculation of Industrial Carbon Emission

Carbon emissions from industrial energy consumption were calculated according to the method provided in the 2006 IPCC National Greenhouse Gas Emission Inventory Guidelines [43]. The calculation method provided by the IPCC is the most widely recognized calculation method in the world. It has the advantage of being highly authoritative and easy to perform. The specific calculation formula is as follows:(1)$$C=\sum_{j}E_{j}\times T_{j}\times F_{j}$$
where C represents carbon emissions. This paper calculates the consumption of raw coal, coke, crude oil, gasoline, kerosene, diesel oil, fuel oil, natural gas, and electric power; $T_{j}$ represents the conversion coefficient of class *J* energy standard coal, and $F_{j}$ is the carbon emission coefficient of the energy. The carbon emission coefficient is provided by the IPCC (2006).

### 3.2. Calculation of Industrial Land Efficiency

ILUE is a reflection of the extent of the industrial land utilization and the corresponding industrial output. Single factor productivity can measure the unit output capacity of the land, which is helpful for evaluating the efficiency of land usage and the dynamic change of land factors [44]. Therefore, this paper uses the industrial output per unit land area to measure the ILUE. This method can directly reflect the relationship between land input and economic output. The specific calculation formula is as follows:(2)$$ILUE_{it}=\frac{TIOV_{it}}{TILA_{it}}$$
where $ILUE_{it}$ refers to the industrial land use efficiency of province *i* in year *t*, $TIOV_{it}$ refers to the total industrial output value of province *i* in year *t*, and $TILA_{it}$ refers to the total industrial land area of province *i* in year *t*.

### 3.3. Panel Threshold Regression Model

The panel threshold model can divide the interval endogenously according to the characteristics of the data itself [35]. The panel threshold models can also study the non-uniform links between explained variables and the explanatory variables. In addition, if the sample segmentation and non-uniform relations between the dependent variables and independent variables are jointly determined, this model can solve problems related to the assumption that sample segmentation is exogenous, which were an issue in previous studies [45]. In this paper, the panel threshold regression model is used to test the threshold effect of industrial land use efficiency on industrial carbon emissions.

The single-threshold model is:(3)$$Y_{it}=\mu_{it}+\beta_{1}X_{it}I(q_{it}\leq \gamma )+\beta_{2}X_{it}I(q_{it}>\gamma )+\varepsilon_{it}$$
where the explained variable *Y* is scalar, μ is the fixed-effect, *X* is a vector of the regressors, *I*(·) is an indicative function, q is the threshold variable, and ε is also scalar.

When the minimum sum of squares of the residuals is $S_{1}(\gamma )$, the optimal estimation of the corresponding threshold is:(4)$$\overset{\wedge }{\gamma }={arg}_{\gamma }minS_{1}(\gamma )$$

The panel threshold model involves two hypothesis tests: (1) Testing whether the threshold effect exists; and (2) Testing whether the estimated threshold value is equal to the true value. The first test and alternative hypotheses are given as follows: (5)$$H_{0}:\beta_{1}=\beta_{2}\;\;H_{1}:\beta_{1}\neq \beta_{2}$$

The statistics are obtained as follows:(6)$$F_{1}=\frac{S_{0}-S_{1}(\overset{\wedge }{\gamma })}{\overset{\wedge }{\sigma^{2}}}$$

If the standard distribution assumption is not satisfied, the bootstrap method is used to obtain the critical value of the approximate distribution. The second test $H_{0}$:$\gamma =\overset{\wedge }{\gamma }$ statistic $LR_{1}$ is obtained as follows:(7)$$LR_{1}=\frac{S_{1}-S_{1}(\overset{\wedge }{\gamma })}{\overset{\wedge }{\sigma^{2}}}$$

For the case of multiple thresholds, the model is set such that:(8)$$Y_{it}=\mu_{i}+\beta_{1}X_{it}I(q_{it}\leq \gamma_{1})+\beta_{2}X_{it}I(\gamma_{1}<q_{it}\leq \gamma_{2})+\beta_{3}X_{it}I(q_{it}>\gamma_{2})+\varepsilon_{it}$$

In order to better verify the nonlinear relationship between industrial land use efficiency and industrial carbon emissions, this paper refers to the panel threshold regression model proposed by Hansen [35] and introduces the per capita GDP and proportion of R&D expenditure as threshold variables. Considering the possible multi-threshold results, the following multi-threshold model is constructed:(9)$$\begin{array}{ll} \ln C_{it} & =\ln\alpha_{it}+\beta_{1}\ln ILUE_{it}+\beta_{2}\ln A_{it}+\beta_{3}\ln T_{it}+\beta_{4}\ln IS_{it}+\beta_{5}\ln EI_{it}+\beta_{6}\ln EC_{it}\\ & +\beta_{7}\ln ILUE_{it}\cdot I(\ln h_{it}\leq \gamma_{1})+\beta_{8}\ln ILUE_{it}\cdot I(\gamma_{1}<\ln h_{it}\leq \gamma_{2})+\beta_{9}\ln ILUE_{it}\cdot I(\ln h_{it}>\gamma_{2})+\ln\varepsilon_{it} \end{array}$$
where $C_{it}$ is the annual industrial carbon dioxide emissions of each province; $A_{it}$ is the per capita GDP; $T_{it}$ is the proportion of R&D expenditure in the main business income of an industry; $IS_{it}$ is the proportion of the main business income of high-tech industry in the total main business income of industries above a designated size; $EI_{it}$ is the energy intensity; and $EC_{it}$ is the energy structure. The sum of the threshold value, $h_{it}$, is the threshold variable, and *I*(·) is the index function. The increase in per capita income improves both the population’s ability and willingness to consume energy and consequently affects carbon emissions. Technological progress can improve energy efficiency and thus reduce carbon emissions. The transformation of industrial structures can shift resources to low-carbon, high-tech enterprises and reduce carbon emissions as a result. Becoming less energy-intensive leads to improvements in energy efficiency. Despite other factors remaining unchanged, carbon emissions due to energy utilization will inevitably decrease, and vice versa. Moreover, the growth of carbon emissions is primarily driven by an energy consumption structure dominated by fossil fuels. Therefore, this paper opts to treat an area’s level of economic development, its industrial structure, its level of technological development, energy intensity, and energy structure as control variables. The descriptions of these variables are shown in Table 1.

### 3.4. Spatial Durbin Model

As the efficiency of industrial land use has spatial spillover, we use a spatial econometric model. There are three basic forms of spatial econometric models: the spatial lag model (SAR), the spatial error model (SEM), and the spatial Durbin model (SDM). The spatial Durbin model includes both the spatial lag terms of the dependent variables and the spatial lag term of the error as independent variables, which improves its ability to explain the relationships between spatial variables. Therefore, the spatial Durbin model is established as:(10)$$\ln C_{it}=\alpha +\rho \sum_{j=1}^{N}W_{ij}\ln C_{it}+\phi X_{it}+\theta \sum_{j=1}^{N}W_{ij}X_{it}+c_{i}+\mu_{t}+\varepsilon_{it}$$
where ρ is the impact of local industrial carbon emissions on industrial carbon emissions in neighboring areas; X includes industrial land use efficiency, per capita GDP, proportion of high-tech industries, proportion of R&D expenditure, energy intensity, and energy structure; W is the spatial weight matrix; $c_{i}$ is the individual fixed-effect; $\mu_{t}$ is the time fixed-effect; and $\varepsilon_{it}$ is the random error.

In the spatial Durbin model, the dependent variables of one region potentially affect the dependent variables of other regions. Lesage and Pace [46] argued that using the regression results to directly explain the influence of independent variables on the dependent variables was inaccurate, so the partial differential equation is used to divide the spatial effect into a direct effect and an indirect effect instead:(11)$$\ln C_{it}={\left(1-\rho W\right)}^{-1}+{\left(1-\rho W\right)}^{-1}(X\phi +WX\theta )+{\left(1-\rho W\right)}^{-1}\varepsilon$$

The partial differential equation matrix of the *k*-th variable in the dependent variable is as follows:(12)$$\begin{array}{l} \left[\begin{array}{ccc} \frac{\partial \ln C}{\partial X_{1k}} & \cdots & \frac{\partial \ln C}{\partial X_{nk}} \end{array}\right]=\left[\begin{array}{ccc} \frac{\partial \ln C_{21}}{\partial X_{1k}} & \cdots & \frac{\partial \ln C_{2n}}{\partial X_{nk}}\\ \cdots & \cdots & \cdots \\ \frac{\partial \ln C_{2n}}{\partial X_{1k}} & \cdots & \frac{\partial \ln C_{2n}}{\partial X_{nk}} \end{array}\right]\\ ={\left(1-\rho W\right)}^{-1}\left[\begin{array}{cccc} \varphi_{k} & W_{12}\theta_{k} & \cdots & W_{1n}\theta_{k}\\ W_{21}\theta_{k} & \varphi_{k} & \cdots & W_{2n}\theta_{k}\\ \cdots & \cdots & \cdots & \cdots \\ W_{n1}\theta_{k} & W_{n2}\theta_{k} & \cdots & \varphi_{k} \end{array}\right] \end{array}$$

The average values of diagonal and off-diagonal elements represent direct and indirect effects, respectively, and the sum of the matrices is the total effect.

## 4. Data Source

This study selected 30 provinces, autonomous regions and municipalities directly under the central government (excluding Hong Kong, Macao, Taiwan and Tibet) as the research objects. The initial research year was 2003. The data on carbon emissions were calculated according to the formula in the Intergovernmental Panel on Climate Change (IPCC, 2006). The data on industrial land efficiency were collected from the China Urban Statistical Yearbook (2004–2019), the China Urban Construction Statistical Yearbook (2003–2018), Statistical Bulletins of national economic and social development of local cities and Statistical Yearbooks of local cities. The data on energy intensity and energy resource structures come from the China Energy Statistical Yearbook (2004–2019). The data on per capita GDP and technological development levels come from the China Statistical Yearbook (2004–2019). The data on industrial structures come from the China High Tech Industry Statistical Yearbook (2004–2019).

## 5. Results and Discussion

### 5.1. Spatiotemporal Distribution Characteristics of Industrial Land Use Efficiency

To more intuitively show the spatial distribution of ILUE in China’s provinces, this paper uses the ArcGIS software platform. The study analyzes the spatial distribution of ILUE in four stages, namely during 2003, 2008, 2013, and 2018. As shown in Figure 2, the spatial distributions of ILUE in different provinces of China are noticeably different, but the intensity of the change trends during the four time periods remains small. There are few provinces with high level of ILUE, and the distribution difference is obvious, mainly in the eastern coastal regions. From 2003 to 2013, the ILUE of China’s provinces displayed a trend of overall improvement, with little difference in spatial distribution, showing a pattern of high ILUE in the eastern region and low ILUE in the western region. In 2013 and 2018, the overall trend was similar, and the ILUE in some provinces began to decline. In the Midwest, only Qinghai province had a high level of ILUE. The data on industrial land use efficiency was an aggregation of data from prefecture-level cities in each province, chosen for this study primarily due to their availability. There is only one prefecture-level city in Qinghai province, and other provinces with high levels of ILUE are concentrated in the eastern region. In general, ILUE in the eastern region is higher than that in the central and western regions. Due to its geographical advantages, the eastern region enjoyed early industrial development, the rapid upgrading and transformation of its industrial structures, and the strong presence of industrial R&D.

### 5.2. Regression Results of Threshold Model

In order to investigate the nonlinear relationship between ILUE and industrial carbon emissions, this study tested whether there were thresholds before conducting the regression. We chose to include the level of per capita GDP (A) and technological development level (T) in the model as the threshold variables. The study used “self-sampling” to obtain the *p*-value and to determine the number of thresholds. As shown in Table 2, Table 3 and Table 4, in the eastern region, both the A and T threshold variables had a single-threshold effect at the 1% level and the 5% level. The model of the central region revealed that the threshold variable A had a double-threshold effect, with a significance level of 10%. However, the model of the western region indicated no thresholds.

As shown in Table 5, the effect of ILUE on industrial carbon emissions was different in the eastern, central, and western regions. When A was taken as the threshold variable, the ILUE had a single-threshold effect on industrial carbon emissions in the eastern region. Therefore, a single-threshold model was established for the regression analysis. The threshold value A was 49,996.09 yuan for industrial carbon emissions. In different A intervals, the influence of ILUE on industrial carbon emissions differed, showing significant nonlinear characteristics. When A was lower than 49,996.09 yuan, ILUE had a remarkably positive influence on industrial carbon emissions (the coefficient was 0.3152). When A was greater than 49,996.09 yuan, the influence on ILUE was positive (the coefficient was 0.2747), but the positive effect was significantly reduced. The results showed that after A reached a certain level, the positive effect of ILUE on industrial carbon emissions was reduced. This illustrated that, although the eastern region was economically developed, there were also some problems in its development, such as unreasonable industrial structures and serious industrial pollution. As for the other control variables, A, T, and EI had a promotional effect on industrial carbon emissions. This is consistent with the results of related studies [47]. Economic growth is often accompanied by rapid energy consumption, resulting in an increase in carbon emissions. Energy intensity is the energy consumption per unit of GDP, that is, the greater the energy intensity, the more carbon emissions. The possible reason is that there are still heavy industrial projects with high energy consumption in the eastern region, which is not conducive to carbon emission reduction. IS and EC are opposites, which was also verified in other relevant studies [48]. The increase in the proportion of high-tech industries means that the mode of production is more intensive and cleaner, which is conducive to reducing carbon emissions.

When T was taken as the threshold variable, ILUE had a single-threshold effect on industrial carbon emissions in the eastern region. In this case, a single-threshold model was used for the analysis. As shown in Table 5, the threshold value of T was 0.0014 for industrial carbon emissions. In different T intervals, the impact of ILUE on industrial carbon emissions differed, showing significant nonlinear characteristics. When T was less than 0.0014, the influence of ILUE on industrial carbon emissions was both positive and significant (the coefficient was 0.3786). When T was greater than 0.0014, the influence on ILUE was positive and significant (the coefficient was 0.3285), but the positive effect was significantly reduced. The results showed that after T reached a certain level, the positive influence of ILUE on industrial carbon emissions was reduced. This is contrary to the negative result we expected. A possible reason for this is the rebound effect of technology. Technological progress can reduce carbon emissions, but it is also accompanied by greater energy consumption [49]. The rebound effect of the environment can partially or completely offset the expected reduction in carbon emissions, so that the final impact coefficient is still positive. This is consistent with the conclusion of Wang et al. [50].

Taking A as the threshold variable, in central China, ILUE showed a double-threshold effect on industrial carbon emissions. Based on this finding, the double-threshold model was employed to carry out the regression analysis. As shown in Table 5, the industrial carbon emission threshold A is 11,833.62 yuan and 12,081.13 yuan, respectively. When A is in a different interval, the impact of ILUE on industrial carbon emissions differs, showing significant nonlinear characteristics. When A is less than 11,833.62 yuan, the influence of ILUE on industrial carbon emissions is significant and positive (the coefficient is 0.4175). When A is greater than or equal to 11,833.62 yuan and less than 12,081.13 yuan, the influence of ILUE on industrial carbon emissions is significant and positive (the coefficient is 0.4040), and the positive effect is slightly reduced. When A is greater than 12,081.13 yuan, the impact coefficient is 0.4143, and the promotion effect is enhanced. Therefore, the increase in ILUE is not beneficial to the environment in the central region. Although the economy has transitioned through different stages, the harmful impacts of relying on heavy industry have not been alleviated.

### 5.3. Regression Results of Spatial Durbin Model

#### 5.3.1. Estimation Results of Spatial Panel Durbin Model

This paper used a 0–1 geographical proximity matrix as the spatial weight matrix, and constructed SDM for the relationship between ILUE and industrial carbon emissions. The 0–1 geographical proximity matrix can accurately describe the geographical relationships between provinces. The results are shown in Table 6. Comparing the effects of the three models of space, time, and a double fixed-effect model of space and time, it was found that the double fixed-effect model of space and time performed better than the other two models. Therefore, we employed the double fixed-effect model of space and time to examine the impacts of ILUE on carbon emissions.

To allow an in-depth analysis, this study decomposed the spatial spillover effect. As shown in Table 7, the total effect can be divided into two parts, one of which is the direct effect, which represents the impact of ILUE on local carbon emissions. The other part is the indirect effect, that is, the spillover effect, which represents the influence of local ILUE on carbon emissions in areas using a spatial correlation. Under the 0–1 geographical proximity weight matrix, the indirect spillover effect of ILUE on reducing regional carbon emissions was significant, and the indirect effect was even greater than that on local carbon emissions.

The original effect coefficient and the direct effect coefficient of ILUE are 0.0587 and 0.0662, respectively, and both pass the 1% significance test, indicating that the promotion effect is statistically significant. At the same time, the spatial effect coefficient and indirect effect coefficient of ILUE are −0.0857 and −0.0861, respectively. The former means that increasing ILUE in neighboring provinces reduces carbon emission levels in a province, while the latter refers to the province where ILUE increases reduce the carbon emission levels of ILUE in neighboring provinces. The former passes a significance test of 10%, and the latter passes a significance test of 5%. After the industrial land efficiency of each province is improved, this more intensive industrial land use mode can promote the carbon emission effect of adjacent provinces. On the one hand, it reduces the radiation of carbon emissions to neighboring areas through the industrial transmission mechanism. On the other hand, neighboring areas improve efficiency by learning advanced measures of intensive use of industrial land, so as to realize carbon emission reduction. This shows that each province involved in the process of improving ILUE can alter its own carbon emission effects while having a restraining effect on the carbon emissions of neighboring provinces. The total effect of ILUE on carbon emission levels is −0.0199, but the result is non-significant. This stems from the on-going development of China, which remains in an intermediate period of industrial development and still suffers serious problems with the extensive use of industrial land. Consequently, improvements in ILUE do not fully take into account the environmental impacts. Therefore, while ILUE does have a certain degree of impact on carbon emission levels, it does not have a strong effect.

A had a significant promotional effect on the direct impact of industrial carbon emissions. China is in a mid- to late-stage of industrial development. The main driving force behind economic growth is energy consumption, which intensifies carbon emissions [51]. The indirect effect of A is positive but non-significant, which indicates that the transmission of economic growth to neighboring provinces is a long-term process, and the effect is non-significant in the short term. Based on its overall effect, the influence of A on industrial carbon emissions is positive. This is also largely consistent with the research results of Ren et al. [52]. The direct effect of T on industrial carbon emissions is positive but non-significant, indicating that technological innovation with preferences for a rebound effect do not play a role in reducing carbon emissions. The indirect effect of the ratio of T to industrial carbon emissions is −0.1415, indicating that the improvement of technological development levels in a province can inhibit the carbon emissions of neighboring provinces. The production process is relatively advanced, with a high utilization rate. Through learning and exchange, all of the provinces have improved their production technologies and reduced carbon emissions. From the overall effect, for a long time, the influence of technological progress of each province on the level of carbon emissions has a certain inhibitory effect [53]. IS and EI play significant parts in promoting carbon emissions. In other words, the high energy consumption of economic growth, extensive energy use, and ineffective industrial structures have not been effectively addressed and controlled [54].

#### 5.3.2. Estimation Results of Regional Panel Durbin Model

As shown in Table 8, SDM was used to analyze the impact of ILUE on industrial carbon emissions in different regions. The spatial spillover effect was non-significant in the eastern region, while the spatial spillover effect was significant in the central and western regions. Except for the central region, the three effects of ILUE on industrial carbon emissions were non-significant. Specifically, the direct effect coefficient and the indirect effect coefficient of ILUE were 0.2288 and 0.1247, respectively. The former illustrates that the increase in ILUE in the local province increased its carbon emission level, while the latter indicates that the increase in ILUE in the local province increased the carbon emission levels of ILUE in neighboring provinces. This showed that each province had a promotional influence on both its own the carbon emissions and those of surrounding provinces. The total effect of ILUE on carbon emission levels was positive (0.3535) at the significance level of 1%. A possible reason for this is that the industrial structure of the central region is still dominated by heavy industry. Although ILUE was improved, the problematic industrial structure led to the increase in industrial carbon emissions. At the same time, there was a spatial spillover effect in the central region. The increase in ILUE in one region may cause the industrial carbon emissions of that region to flow into the adjacent regions through high-carbon industries and emissions, resulting in increased carbon emissions in the adjacent regions.

## 6. Conclusions and Policy Implications

This paper studies the impacts of industrial land use efficiency on the carbon emissions of 30 provinces in China. The panel threshold regression model was used to test the threshold effect of industrial land use efficiency on industrial carbon emissions. The spatial Durbin model was used for the in-depth analysis of ILUE on the spatial spillover effects of carbon emissions. The results showed that the overall ILUE in China was higher in the eastern region and lower in the western region. The threshold effects of T in the central and western regions were non-significant, which indicates that the overall technological development levels of the central and western regions were low. The spatial spillover effect of ILUE on industrial carbon emissions is non-significant in China.

The above conclusions have value as references for the government’s efforts to formulate reasonable industrial policies.

Firstly, the government should ensure industrial parks are well-planned, work to promote the good layout of industrial spaces, and make full use of industrial sites to improve the utilization efficiency of industrial land. If utilization efficiency improves, pollution and emissions will be reduced. At different stages of economic development, the government should take targeted measures to improve ILUE in specific regions. For the eastern region, where the economy is at an advanced stage of development, the results indicate that the current industrial structure promotes carbon emission reductions, indicating that the proportion of high-tech industries in the eastern region is high, and transforming and upgrading the industrial structure is reasonable. However, increases in ILUE in the eastern region still have positive effects on industrial carbon emissions. The local government should continue to develop clean energy and clean technologies to reduce their industrial carbon emissions. For the central and western regions, where the economies are at an early stage of development, there may also be a model of growth based on “promoting development by land”, which may carry the cost of higher carbon emissions. In light of this, the government should strengthen the examination and approval standards of industrial land, set a high access threshold for industrial enterprises, carefully plan the use of industrial parks, vigorously support the entry of intensive industrial enterprises, and avoid the waste of idle industrial land. Moreover, the government should increase investment in technological innovation in the central and western regions. We suggest increasing the incentive for industrial enterprises to innovate, in order to develop more efficient production technologies, innovative abilities and management methods, and to promote the improvement of industrial green production efficiency and environmentally friendly qualities.

Secondly, supporting policies for industrial land supply should also be taken into consideration, including increasing the weight of green development in performance appraisals, strengthening environmental regulations, pursuing innovation in green financial policies, etc. For example, a big data platform based on a comprehensive informational survey of the economy, society, and the environment of industrial land could provide basic information for optimizing decision-making and research processes involving the industrial land supply.

The conclusion has provided relevant reference for the government to improve the efficiency of industrial land use and reduce carbon emissions in China. However, the research is still preliminary. This paper uses single factor productivity to calculate industrial land efficiency, without considering the impact of other production factors on output. Besides, since industries include different sectors, the impact of industrial land efficiency on carbon emissions may be different among different sectors. However, this paper does not take into account the differences.

## Figures and Tables

**Figure 1 ijerph-18-09368-f001:**
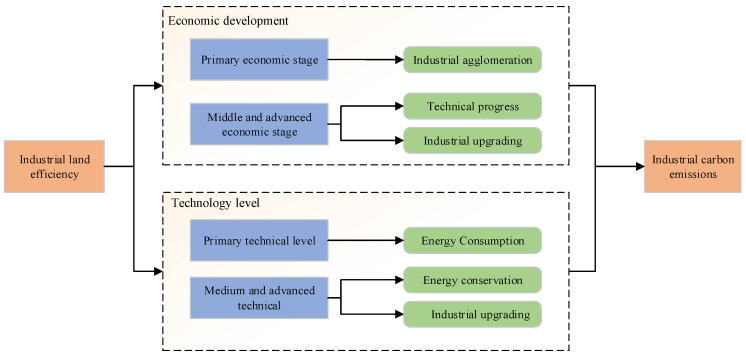
Threshold mechanism of industrial land use efficiency.

**Figure 2 ijerph-18-09368-f002:**
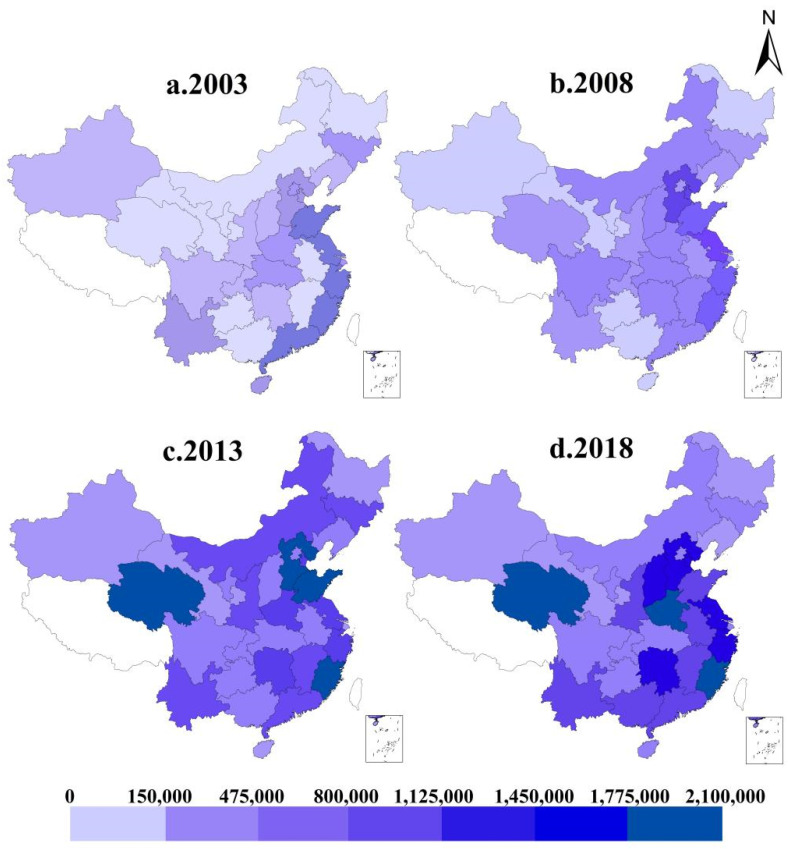
Changes of industrial land use efficiency in 2003, 2008, 2013 and 2018.

**Table 1 ijerph-18-09368-t001:** Description of variables.

Variable	Definition	Unit
Industrial carbon emissions (C)	Carbon emissions from the industrial sector	10^4^ t
Industrial land use efficiency (ILUE)	Industrial output value per unit industrial land of province i	10^4^ yuan/km^2^
Economic development level (A)	Per capita GDP	yuan
Technical level (T)	Proportion of R&D expenditure in main business income of Industrial Enterprises above Designated Size	%
Industrial structure (IS)	Proportion of main business income of high-tech industry in main business income of Industrial Enterprises above Designated Size	%
Energy intensity (EI)	Proportion of industrial standard coal consumption in industrial added value	t/10^4^ yuan
Energy structure (EC)	Proportion of industrial coal consumption standard coal in total consumption standard coal	%

**Table 2 ijerph-18-09368-t002:** Tests for threshold effect and threshold value estimation of the eastern region.

Threshold Variable	Number of Thresholds	F-Statistic	Threshold Value	95% Confidence Interval
lnA	Single	50.31 ***	10.8197	(10.7910, 10.8292)
	Double	10.72	10.8197	(10.7910, 10.8292)
			9.6039	(9.5850, 9.6182)
	Triple	13.03	9.8974	(9.8803, 9.9027)
lnT	Single	28.92 **	−6.5548	(−6.9425, −6.3814)
	Double	9.11	−6.5548	(−6.9425, −6.3814)
			−5.1616	(−5.1924, −5.1552)
	Triple	7.94	−5.1720	(−5.2055, −5.1584)

Note: *** and ** represent significance level of at 10% and 5% respectively.

**Table 3 ijerph-18-09368-t003:** Tests for threshold effect and threshold value estimation of the central region.

Threshold Variable	Number of Thresholds	F-Statistic	Threshold Value	95% Confidence Interval
lnA	Single	9.84	9.2997	(9.2867, 9.3037)
	Double	14.80 *	9.3994	(9.3959, 9.3995)
			9.3787	(9.3759, 9.3840)
	Triple	3.55	9.3871	(9.3855, 9.3900)
lnT	Single	13.00	−4.7340	(−4.7682, −4.6823)
	Double	8.41	−4.7340	(−4.7674, −4.6823)
			−5.2701	(−5.3749, −5.2661)
	Triple	6.36	−5.9458	

Note: * represent significance level of at 1%.

**Table 4 ijerph-18-09368-t004:** Tests for threshold effect and threshold value estimation of the western region.

Threshold Variable	Number of Thresholds	F-Statistic	Threshold Value	95% Confidence Interval
lnA	Single	16.94	8.4129	(8.3394, 8.5996)
	Double	13.78	8.4129	(8.3531, 8.5996)
			9.1698	(9.1590, 9.1783)
	Triple	5.04	9.6093	(9.5847, 9.6183)
lnT	Single	5.84	−5.6743	(−5.6899, −5.6668)
	Double	3.99	−5.6743	(−5.6899, −5.6668)
			−4.5932	
	Triple	4.00	−5.4743	(−5.4806, −5.4655)

**Table 5 ijerph-18-09368-t005:** Threshold A regression results in the eastern, central, and western regions.

Eastern Region				Central Region	
Variable	A	Variable	T	Variable	A
lnA	1.3927 ***	lnA	1.6263 ***	lnA	1.0152 ***
	(8.08)		(8.69)		(8.29)
lnT	0.1492 ***	lnT	0.3263 ***	lnT	−0.0477
	(4.45)		(7.31)		(−1.09)
lnEC	−0.4065 ***	lnEC	−0.2264 ***	lnEC	0.2995 ***
	(−8.06)		(−4.39)		(6.14)
lnEI	0.6077 ***	lnEI	0.8417 ***	lnEI	0.3516 ***
	(11.14)		(14.17)		(8.00)
lnIS	−0.1288 *	lnIS	−0.1085	lnIS	0.0019
	(−1.91)		(−1.53)		(0.06)
lnILUE (q1 < 10.8197)	0.3152 ***	lnILUE (q1 < −6.5548)	0.3786 ***	lnILUE (q1 < 9.3787)	0.4175 ***
	(6.35)		(7.32)		(12.00)
lnILUE (q1 ≥ 10.8197)	0.2747 ***	lnILUE (q1 ≥ −6.5548)	0.3285 ***	lnILUE (9.3787 ≤ q1 < 9.3994)	0.4040 ***
	(5.42)		(6.34)		(11.63)
				lnILUE (q1 ≥ 9.3994)	0.4144 ***
					(11.68)
_cons	−9.6990 ***	_cons	−11.0256 ***	_cons	−6.6708 ***
	(−6.54)		(−7.01)		(−7.52)

Note: ***and * represent significance level of at 10% and 1%, respectively.

**Table 6 ijerph-18-09368-t006:** Results of Durbin regression.

Variables	Individual Fixed	Time Fixed	Both
lnILUE	0.1469 ***	0.6582 ***	0.0587 ***
lnA	0.9940 ***	0.1648 *	1.0400 ***
lnT	0.0748 ***	0.2141 ***	0.0223
lnEC	−0.0861 **	0.4058 ***	−0.0465
lnEI	0.6384 ***	−0.0275	0.6537 ***
lnIS	0.0461 *	−0.1361 ***	0.0302
WlnILUE	0.1803 ***	0.8314 ***	−0.0857 *
WlnA	0.1978	−0.3430 *	0.5289 ***
WlnT	0.0254	−0.1607	−0.1659 ***
WlnEC	−0.0860	0.8464 ***	−0.0864
WlnEI	−0.0388	0.0685	0.3116 ***
WlnIS	0.0768 **	−0.3361 ***	0.1127 **
Spatial rho	0.1788 ***	−0.0305	−0.2991 ***
R2	0.8537 290.0283 0.0174 *** 480	0.5986 −380.8563 0.2859 *** 480	0.0447 365.5730 0.0126 *** 480
Log-L			
sigma2			
obs			

Note: ***, ** and * represent significance level of at 10%, 5% and 1%, respectively.

**Table 7 ijerph-18-09368-t007:** Spatial Durbin indirect effect and direct effect.

Variable	Both		
	Direct Effect	Indirect Effect	Total Effect
lnILUE	0.0662 ***	−0.0861 **	−0.0199
lnA	1.0212 ***	0.1808	1.2019 ***
lnT	0.0359	−0.1415 ***	−0.1056 **
lnEC	−0.0424	−0.0579	−0.1003 **
lnEI	0.6457 ***	0.0951 **	0.7408 ***
lnIS	0.0242	0.0895 **	0.1136 ***

Note: *** and ** represent significance level of at 10% and 5% respectively.

**Table 8 ijerph-18-09368-t008:** Spatial Durbin indirect effect and direct effect of eastern, central and western regions.

Variable	Both								
	Eastern Region			Central Region			Western Region		
	Direct Effect	Indirect Effect	Total Effect	Direct Effect	Indirect Effect	Total Effect	Direct Effect	Indirect Effect	Total Effect
lnILUE	0.0782 *	−0.1235 **	−0.0453	0.2288 ***	0.1247 *	0.3535 ***	0.0361	0.0114	0.0475
lnA	0.9590 ***	0.6768 ***	1.6359 ***	0.9480 ***	0.1294	1.0775 ***	0.6405 ***	−0.3724	0.2681
lnT	0.0229	0.1936 ***	0.2165 ***	−0.2342 ***	−0.0116	−0.2458 *	0.0793 *	−0.1544 *	−0.0751
lnEC	−0.2984 ***	−0.0128	−0.3112 ***	0.0734	−0.2734 ***	−0.2000 **	−0.1053 *	−0.0323	−0.1376
lnEI	0.8489 ***	−0.3903 ***	0.4585 ***	0.3719 ***	0.1039	0.4758 ***	0.6310 ***	0.0272	0.6583 ***
lnIS	−0.0189	−0.0016	−0.0205	−0.0640	−0.0731	−0.1371 **	0.0223	−0.1493 ***	−0.1269 **
Spatial rho	0.0433			−0.1849 **			−0.3529 ***		

Note: ***, ** and * represent significance level of at 10%, 5% and 1%, respectively.

## Data Availability

The datasets used and analyzed during the current study are available from the correspondent author on reasonable request.

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
