# Peer review of "Spatial Threshold Effect of Industrial Land Use Efficiency on Industrial Carbon Emissions: A Case Study in China"

_ijerph, 2021, doi:10.3390/ijerph18179368_

Round 1
Reviewer 1 Report
Dear authors,
Thank you for your invitation.
This paper tries to identify and solve in important question. The methodology is sound. However, the paper provides too many pieces of information without effectively provide a reasonable link among them and explain how this information could be used in answering the research question and for better land use and industrial carbon emission for the future of the study region. I have some major concerns listed below:
- Introduction: The author should put forward the research purpose of this article before the statement “The remainder of this paper is organized as follows.”
- Literature review: In general, I found that this section has some very generic statements and the issue discussed in the paper is not seen in the perspective of other works that have dealt with issues of Industrial carbon emissions; industrial land efficiency; threshold-model etc.
- Methods: what is the justification of adopting Calculation of industrial carbon emission, Calculation of industrial land efficiency, Panel threshold regression model? The format of equation 9 and table 1 should be improved. How come we get to variables in Table 1?
- Data source: “This study selected 30 provinces, autonomous regions and municipalities of China as the research subjects except Tibet.” Was Taiwan province also included the research subjects? If so, why there is no data of Taiwan province in Fig.2?
- Results and discussion: some interesting insights but still lack of coherency with the introduction/literature. still simply copy paste from commissioned works.
- Conclusion and policy implications: it mostly looks like a summary of the work done and the results obtained. Also, there should be some content in the conclusion regarding the limitations of the current research and future work possibilities.
- There are more language and grammar issues than mentioned above. I highly recommend using a professional editor to correct them before re-submission.
- Some literature to consult:
An approach to examining performances of cool/hot sources in mitigating/enhancing land surface temperature under different temperature backgrounds based on landsat 8 image. Sustainable Cities and Society, 44, 416-427.
Demand prediction and regulation zoning of urban-industrial land: Evidence from Beijing-Tianjin-Hebei Urban Agglomeration, China. Environ Monit Assess 191, 412 (2019).
Understanding land surface temperature impact factors based on local climate zones, Sustainable Cities and Society (2021)
Author Response
Comment 1: Introduction: The author should put forward the research purpose of this article before the statement “The remainder of this paper is organized as follows.”
Response: Thank you for your valuable advice. And we have supplemented the research purpose in the Introduction.
“This study aims to assess China’s industrial land use efficiency, investigate spatiotemporal distribution characteristics of industrial land use efficiency, and analyze the spatial threshold effect of industrial land use efficiency on industrial carbon emission. The conclusions have important reference value for optimizing industrial layout and realizing carbon emission reduction.” (Line 82-86)
Comment 2: Literature review: In general, I found that this section has some very generic statements and the issue discussed in the paper is not seen in the perspective of other works that have dealt with issues of Industrial carbon emissions; industrial land efficiency; threshold-model etc.
Response: Thank you for your valuable advice. We have completely revised the literature review to ensure that it is consistent with our work.
Comment 3: Methods: what is the justification of adopting Calculation of industrial carbon emission, Calculation of industrial land efficiency, Panel threshold regression model? The format of equation 9 and table 1 should be improved. How come we get to variables in Table 1?
Response: Thank you for your comment. We have added the reasons why we choose these methods in the modified manuscript. We use the calculation method provided by IPCC to calculate industrial carbon emission. It is the most mainstream and most recognized calculation method in the world. It has the advantages of high authority and easy calculation (Line 170-172). ILUE is the reflection of the degree of industrial land utilization and the corresponding industrial output. Single factor productivity can measure the unit output capacity of land, which is helpful to evaluate the use efficiency and dynamic change of land factors. (Chen et al., 2018). Therefore, this paper uses the industrial output per unit land area to measure the ILUE (Line 180-184). The panel threshold models can study the non-uniform links between explained variables and explanatory variables. In addition, if the sample segmentation and non-uniform relations between the dependent variable and independent variables are jointly determined, this model can solve the problems related to the assumption that sample segmentation is exogenous in previous studies (Kuo et al., 2013). In this paper, the panel threshold regression model is used to test the threshold effect of industrial land use efficiency on industrial carbon emission (Line 191-197). The spatial Durbin model includes both the spatial lag terms of the dependent variables and the spatial lag term of the error as independent variables, which has a better ability to explain the relationship between spatial variables. Therefore, the spatial Durbin model is established (Line 239-242).
Comment 4: Data source: “This study selected 30 provinces, autonomous regions and municipalities of China as the research subjects except Tibet.” Was Taiwan province also included the research subjects? If so, why there is no data of Taiwan province in Fig.2?
Response: Thank you for your valuable comment. We are very sorry for the unclear statement in the paper. This study selected 30 provinces, autonomous regions and municipalities directly under the central government (excluding Hong Kong, Macao, Taiwan and Tibet) as the research objects. And we have corrected the description of the study object in 4.Data source.( Line 261-263)
Comment 5: Results and discussion: some interesting insights but still lack of coherency with the introduction/literature. still simply copy paste from commissioned works.
Response: Thank you for your valuable comment. We have gave more explanation for the results and performed a consistency analysis with other literature.(e.g. Line 288-295, Line333-341, Line 353, Line 356, Line 408-413, 423-425.434-436)
Comment 6: Conclusion and policy implications: it mostly looks like a summary of the work done and the results obtained. Also, there should be some content in the conclusion regarding the limitations of the current research and future work possibilities.
Response: Thank you for your comment. We have added the limitations in the last of this paper. Firstly, this paper uses single factor productivity to calculate industrial land efficiency, without considering the impact of other production factors on output. Secondly, since industries include different sectors, the impact of industrial land efficiency on carbon emissions may be different among different sectors. However, this paper does not take into account the differences.(Line 474-480)
Comment 7: There are more language and grammar issues than mentioned above. I highly recommend using a professional editor to correct them before re-submission.
Response: Thank you for your valuable advice. We are very sorry for this language problem. We have try our best to improve the language of this paper. Besides, the manuscript also modified the language by the English Editing Services of MDPI to make sure there is free of grammatical, spelling, and other common errors in our paper.
Comment 8: Some literature to consult:
Response: Thank you for your valuable advice. We have read these literatures and cited them in the modified manuscript.

Reviewer 2 Report
The article "Spatial Threshold Effect of Industrial Land Use Efficiency on Industrial Carbon Emission: A Case Study in China" offers an interesting point of view about development and pollution.
However the abstract does not say anything about the possible practical derivations of the analysis. It can be improved.
The literature review is rich enough but is not well argued in relation to the main focus of the paper, It is a list of other studies. While I suggest you to discuss the most interesting references, explaining the connections with the research development.
Fig. 2 must be revised is blurred and the legend is to small to be read.
The chapter Results and discussion must be improved. I suggest you to describe better results implications, and make in evidence spatial evidence and the analyzed data. Moreover, in this chapter some hypotheses are done but no references are used to support them or to strengthening the discussion.
Conclusion and policy implications must be improved, economic and planning suggestions have to be given.
Finally, I suggest a general revision of typos for ex. see lines 116, 137, 153, etc...
Author Response
Comment 1: However the abstract does not say anything about the possible practical derivations of the analysis. It can be improved.
Response: Thank you for your valuable advice. We have completely revised the abstract and added the possible practical derivations of the analysis in the last.
Comment 2: The literature review is rich enough but is not well argued in relation to the main focus of the paper, It is a list of other studies. While I suggest you to discuss the most interesting references, explaining the connections with the research development.
Response: Thank you for your valuable advice. We have completely revised the literature review to ensure that it is consistent with our work.
Comment 3: Fig. 2 must be revised is blurred and the legend is to small to be read.
Response: Thank you for your valuable advice. And we have modified the clarity and the legend of Figure 2.
Comment 4: The chapter Results and discussion must be improved. I suggest you to describe better results implications, and make in evidence spatial evidence and the analyzed data. Moreover, in this chapter some hypotheses are done but no references are used to support them or to strengthening the discussion.
Response: Thank you for your valuable comment. We have gave more explanation for the results and performed a consistency analysis with other literature.(e.g. Line 286-293, Line332-340, Line 353, Line 355, Line 407-412, 422-424.433-435)
Comment 5: Conclusion and policy implications must be improved, economic and planning suggestions have to be given.
Response: Thank you for your valuable advice. We have completely revised the policy implications and added economic and planning suggestions.
“Secondly, Secondly, supporting policies for industrial land supply should also be taken into consideration, including increasing the weight of green development in performance appraisals, strengthening environmental regulations, pursuing innovation in green financial policies, etc. For example, a big data platform based on a comprehensive informational survey of the economy, society, and the environment of industrial land could provide basic information for optimizing decision-making and research processes involving the industrial land supply.”
Comment 6: Finally, I suggest a general revision of typos for ex. see lines 116, 137, 153, etc...
Response: Thank you for your valuable advice. We have rechecked the original text and corrected these problems.

Reviewer 3 Report
some statements especially quoting values must be referenced.
Some explanations need to be justified and expanded
Lots of grammar changes required.

Author Response
Comment 1: some statements especially quoting values must be referenced.
Response: Thank you for your valuable advice. We refer to some specific values in the introduction, such as " For the sake of solving the problem of carbon emission reduction, China promised to achieve carbon peak by 2030 and carbon neutrality by 2060 in “Paris Agreement” (Xu et al., 2019)" " The total industrial output value only accounts for about 35% of GDP, while energy consumption accounts for nearly 70%, and carbon emission exceeds 80% (Wang and Feng, 2018)". Besides, some explanations in the empirical discussion also have been cited references.
Comment 2: Some explanations need to be justified and expanded
Response: Thank you for your valuable comment. We have gave more explanation for the results and performed a consistency analysis with other literature.(e.g. Line 286-293, Line332-340, Line 353, Line 355, Line 407-412, 422-424.433-435)
Comment 3: Lots of grammar changes required.
Response: Thank you for your valuable advice. We are very sorry for this language problem. We have try our best to improve the language of this paper. Besides, the manuscript also modified the language by the English Editing Services of MDPI to make sure there is free of grammatical, spelling, and other common errors in our paper.
Besides, we have revised all the notes marked by the reviewer in the text.

Round 2
Reviewer 1 Report
The revised version is fine.
Reviewer 2 Report
The paper is improved and the reading clear.
Thanks for your revision